# Nutrition, a Tenet of Lifestyle Medicine but Not Medicine?

**DOI:** 10.3390/ijerph18115974

**Published:** 2021-06-02

**Authors:** Leigh A. Frame

**Affiliations:** The George Washington School of Medicine and Health Sciences, Washington, DC 20037, USA; leighframe@gwu.edu

**Keywords:** nutrition, lifestyle medicine, education, medical education, lifestyle, integrative medicine, public health

## Abstract

Nutrition is a foundation of health and one of six pillars of Lifestyle Medicine. The importance of nutrition in clinical care is now widely recognized by health care professionals and the public. However, clinicians are not comfortable counselling their patients on nutrition due to inadequate or lack of training, leaving a significant need in patient care. This gap can be closed with evidence-based curricula in medical schools and in the trainings of other health care professionals. This communication presents the current state of nutrition knowledge in health care, emphasizing nutrition education for physicians, and presents a model of how pre- through post-professional health care providers may become proficient in nutrition counseling including appropriate referral to more specialized providers. With these skills, health care professionals will be able to initiate patient-centered lifestyle plans. This includes improving diet and utilization of team-based medicine and referrals.

## 1. Introduction

As the introduction to the Special Issue, “The Lifestyle Medicine Movement: An Extention of Public Health into Medicine,” this will set the scene for the current state of the Lifestyle Medicine movement using the pillar of nutrition as an exemplar. A brief background will be followed by a case study of a recently designed interprofessional education program for nutrition in health care broadly—no longer confined solely to consultations with Registered Dietitians (RDs), if a patient can be so lucky as to be able to meet with an RD due to the many barriers including lack of referral by clinicians. This, which may serve as a model for other organizations and, thus, increase the preparedness of health care professionals to provide first-line nutrition counseling as well as to refer to more specialized health care providers as appropriate. This article aims to highlight (1) that the Lifestyle Medicine movement is a sign of the poor nutrition training currently provided for health care providers outside of RDs in the United States (US); (2) that this calls for improved clinician training, including and especially within US medical schools; and (3) one potential solution for improved nutrition training for health care professionals.

## 2. Background

Lifestyle is linked to the leading causes of death globally, including overconsumption of alcohol, use of tobacco, sedentary behavior, and poor dietary choices [1]. Despite the growing recognition of the importance of lifestyle factors in prevention of disease and promotion of health and wellness, including their inclusion into practice guidelines [2,3], many patients have not been receiving counselling on this topic, sparking the Lifestyle Medicine movement. 

Nutrition and other lifestyle interventions are considered first-line therapy for many health issues. In fact, 60% of American adults have one or more diet-related chronic diseases [4]. However, 60% are not receiving nutrition counseling [4]. Rather, nutrition counseling is included in an estimated 25–40% of primary care visits [5,6,7,8]—where patients are often presenting for first-line therapy.

In the US, the American College of Lifestyle Medicine (ACLM) defines Lifestyle Medicine as “the use of a whole food, plant-predominant dietary lifestyle, regular physical activity, restorative sleep, stress management, avoidance of risky substances and positive social connection as a primary therapeutic modality for treatment and reversal of chronic disease” [9]. This is a whole person approach to move beyond sick care and foster optimal health through supporting patient behavior change using public health recommendations. The American College of Preventative Medicine (ACPM) has adopted a similar definition: “a medical approach that uses evidence-based behavioral interventions to treat and manage chronic diseases related to lifestyle…[and] more effectively coach patients about nutrition, physical activity, stress management, sleep, social support and environmental exposures” [10]. ACLM and ACPM jointly published a consensus report on the Lifestyle Medicine core competencies [11].

## 3. Nutrition Education & Health Care

It is clear from these definitions that nutrition and physical activity are at the foundation of Lifestyle Medicine. While a whole person approach is preferable, the basics of nutrition are one of the chief tenants—one that is all but missing from health care provider training currently—outside of RDs. Lack of nutrition education for health care professionals affects clinical outcomes; quality of life; quality, utilization, and cost of health care; patient and provider satisfaction [5,6,7,8,11,12].

The importance of nutrition is now widely recognized by health care professionals and the public. A 2016 systematic review of general knowledge in adults across five continents by Barbosa et al. found that greater nutrition knowledge correlated with general well-being as well as socioeconomic factors, especially educational attainment, which may affect both the access to care and healthy foods as well as the knowledge and ability to implement better nutrition [12]. In fact, Barbosa et al. noted that nutrition knowledge is correlated with “healthy food choices and habits” [12].

Despite recognizing the importance of nutrition in chronic diseases such as obesity and cardiovascular disease, clinicians infrequently counsel their patients on nutrition. This is largely due to a knowledge gap.

In 2019, JAMA Internal Medicine published an opinion article by Dr. Neal Barnard entitled, “Ignorance of Nutrition is No Longer Defensible” [13]. Dr. Barnard describes a case in which a patient is hospitalized with severe complications from diabetes, yet there is no mention of nutrition—the root cause of the disease and related comorbidities and complications—even upon discharge. Is it okay that no clinician thought to help this patient through dietary and lifestyle changes (directly or indirectly by referring them to an RD)? Dr. Barnard and I would both argue that this is no longer acceptable even though the majority of clinicians report poor or no nutrition training [6,8,13,14]—61% in one survey of internal medicine residents [15]. A multinational systematic review of qualitative and quantitative studies concluded that nutrition is poorly incorporated into medical education at best no matter the location or student seniority despite student desire for nutrition education [16]. Further, medical students feel unprepared to manage cases where nutrition may potentially affect patient outcomes, which persists once in practice [16].

### 3.1. The State of Nutrition Education for Physicians

The literature supports our understanding that health care providers receive little to no nutrition education and that this is a significant need in patient care. Further, the literature shows that the understandings health care providers have currently may be inaccurate. In a 2018 survey of University of Florida physicians (92%), only 25% knew the American Heart Association recommendation for vegetable and fruit intake while 46% knew the recommendation for physical activity [14]. While neither statistic is comforting, the meager 1 in 4 practicing physicians with such fundamental nutrition knowledge is alarming. When getting into more detailed nutrition recommendations, the results were even less impressive: only 20% new the recommended daily limit for sugar [14].

The nutrition knowledge of these physicians also correlated with their personal behaviors, as one would predict from Step 1 in Thomas et al., *Curriculum development for medical education: A six-step approach* [17], which is typically used in US medical schools. Among those who felt nutrition was important, 40% reported eating at least 2 vegetable and 3 fruit servings daily while only 7% of those who rated nutrition as “neutral,” “not important,” or “important, but I don’t have the time to focus on it right now” (*p* < 0.0001) [14]. Thus, nutrition education for health care providers is likely crucial for the health of these providers as well as their patients.

This is a growing area of research and interest in health care and education of health care providers. While the National Academy of Sciences recommends 25 h of nutrition education in medical schools and the American Society for Nutrition recommends 44 h, the current average is less than 20 h [18]. Further, many schools relegate nutrition education to electives, which are only taken by a self-selecting subset of students, and about half offer any clinical practice session(s) in nutrition. A recent update to the findings of Adams et al. [18] shows no improvement in these statistics [19]. This is despite evidence that nutrition education is a strong predictor (perhaps the strongest) of success on the United States Medical Licensing Examination (USMLE) [20,21,22]. So, we know nutrition is important for patient care, it is a predictor of success on the USMLE, and clinicians largely want to learn more about nutrition, but this education is not currently being provided. How do we change the status quo?

### 3.2. Nutrition Education Improvement: Coming to a Consensus

The National Heart, Lung, and Blood Institute (NHLBI) held a workshop to address the “gap in knowledge by convening experts in clinical and academic health professional school” [19]. This workshop included presentations from the National Board of Medical Examiners, the Accreditation Council for Graduate Medical Education, the Liaison Committee on Medical Education, as well as the American Society for Nutrition. The workshop report details the lessons learned, which have led to the development of a framework for competency-based education (rather than knowledge alone). In this framework, Van Horn et al. focuses on “entrustable professional activities (EPAs),” a.k.a. professional duties or charges, and the two means for mastering EPAs: competencies and milestones. The approach outlined by Van Horn et al. [19] is very similar to the six step method of Thomas et al. [17], see Table 1.

Van Horn et al. goes on to assess existing EPAs related to nutrition and only identified one, found in a collaboration of all gastrointestinal societies: *Assess nutritional status and develop and implement nutritional therapies in health and disease*. This is a broad, over-arching EPA but can still be used in curricular design. For instance, this EPA could be used as an objective with the six competencies as sub-objectives.

In separate work to address the nutrition education gap, the International Association for Medical Science Educators (IAMSE) has developed a goal for nutrition education for health care providers: *All graduating medical students will assess nutritional status and manage the clinical encounter to facilitate a personalized nutritional approach for optimal health* [23]. Van Horn et al. became aware of this parallel effort and began a collaboration with the IAMSE, which was expected to publish finalized objectives in 2020. In the meantime, IAMSE published their learning objectives under the following competency domains in a blog post [24]:Micronutrients and macronutrientsEnergy metabolism—calculating basal energy expenditure body compositionNutrition assessment—BMI weight gain/loss, nutrient deficienciesNutrient requirements throughout the lifecycleTaking a diet/physical activity history, prescription for physical activityPatient counseling and motivational interviewingNutrition in health promotion and disease preventionCritical care—enteral and parenteral nutritionReferral to an RD for nutrition consultEvaluating nutritional evidenceOutpatient and inpatient nutrition management—medical nutrition therapy for acute and chronic disease

### 3.3. Nutrition Education Case Study: George Washington Integrative Medicine

#### 3.3.1. Background

What has George Washington School of Medicine and Health Sciences (GW SMHS) done to remedy this nutrition training gap? Provide training in nutrition to clinicians currently in practice as well as future clinicians through the Integrative Medicine (INTM) Graduate Programs, which already emphasized personalized, lifestyle-focused, patient-centered medicine, which many would call Lifestyle Medicine, as one tool in the substantial toolbox of Integrative Medicine (including referral to RDs). The original curriculum provided 2 weeks of foundational nutrition education in the introductory course with reinforcement in several higher level courses. We launched an updated curriculum in Fall 2019 that added two 3 credit hour courses, which delve deep into nutrition and behavioral counseling, to the original INTM coursework. Additionally, a new nutrition concentration was developed offering a total of 17 credit hours of nutrition (including the two 3 credit hour courses added to the original INTM curricula) as part of the Master of Science in Health Sciences in INTM, which requires 36 credit hours for completion. This nutrition concentration is designed for mastery of integrative nutrition counseling and fulfills the educational requirements for the Certified Nutrition Specialist (CNS) credential, which is overseen by the American Nutrition Association. The CNS is a wellness-centered, master’s level nutrition credential emphasizing personalized nutrition with a pathway specifically for physicians.

With the success of the nutrition concentration, we turned our attention to how to best serve the entire health care profession by providing a variety of nutrition education opportunities. We were already offering multiple continuing education opportunities and wanted to build something between these and the master’s in terms of depth and time commitment. We are currently developing the Graduate Certificate in Integrative Nutrition & Lifestyle Medicine, which will require just 15 credit hours for completion. We hope that this will satisfy the busy health care provider without the time or need for a master’s degree and broaden the audience for this important training, especially in the case of practicing physicians, who may be less likely to enroll in a full master’s program. We anticipate the new graduate certificate will launch in Fall 2021.

#### 3.3.2. Competencies

Below are the recommended competencies from Van Horn et al. and their application to the GW INTM nutrition curricula at present:


*1. Perform basic nutritional assessment in the inpatient setting and recognize when patient needs to be fed:*


The INTM nutrition curricula does not emphasize sick care in the inpatient setting, as it is designed as first line therapy in primary care. Instead, we focus on prevention and treatment of early disease. This is in line with the Lifestyle Medicine movement, which emphasizes preventative and wellness care. Further, the traditional RD/RD Nutritionist (RDN) is well qualified to serve in this inpatient role. Therefore, adding more trained individuals to this inpatient pool will not only provide less of an impact (lower need area) but misses the point of the Lifestyle Medicine movement, which is in the outpatient setting as primary care.


*2. Perform basic nutritional assessment in the outpatient setting:*


This is the focus of the INTM nutrition curricula with each class starting with assessment as the foundation of patient care planning. The outpatient setting is where Lifestyle Medicine is designed to function, and where the need is greatest for first line nutritional counseling. After 4 full weeks of patient-provider relationship training in our introductory course, each student is able to work with a patient to develop a patient care plan that will fit into the patient’s life and, thus, likely be successful. They also maintain a strong relationship to iterate on the plan with this patient’s feedback and as their life evolves. Then in Nutrition I: Assessment, Diagnosis, Intervention, the spend an entire 3-credit course from assessment to the initial development of a nutrition care plan; in this course, we use Krause and Mahan’s Food & the Nutrition Care Process (Part I Nutrition Assessment and Part II Nutrition Diagnosis & Intervention). From Nutrition I on, a series of case studies are completed (both formative and summative) and peer review and discussion occurs with each.


*3. Counsel patients on basic public health nutrition issues (e.g., obesity prevention and treatment, hypertension, cardiovascular disease, and diabetes):*


Even the basic nutrition received in our survey course introduces students to the practice of counseling patients on basic public health nutrition issues. In fact, our introductory course has 4 weeks of such counseling focused on obesity and related diseases, which includes drafting their first patient care plan for a patient with class III obesity and a number of severe comorbidities. This is then reinforced and expanded upon throughout the curriculum where much of the nuance of nutrition and various chronic diseases are explored in depth. Our Nutritional Immunology course is an example of taking this notion and moving beyond the basics of public health (see Section 3.4). All of these contribute to the practice of Lifestyle Medicine in the outpatient primary care setting and build upon the assessment skills discussed above.


*4. Recognize fads (nonevidence-based diets/supplements):*


This is a major emphasis of the INTM nutrition curricula. We want our graduates to be at the cutting edge of the evidence-base and not beyond it. Thus, we teach them to perform literature searches, appraise the evidence, and to trust but verify, in general and especially for new fads, which is especially important in nutrition. The critical thinking and clinical research training they receive are at the heart of this. While each course requires that they support all statements with a citation from the literature (or state that this is anecdotal), we have an entire 3-credit course devoted to Clinical Research in Integrative Medicine (CRIM). In CRIM, each student performs a literature review and develops a Specific Aims page in the style of the National Institutes of Health (NIH). They also undergo NIH training including Good Clinical Practice. Finally, they complete the Publons Academy to learn the core competencies of peer review and become Certified Peer Reviewers. This is crucial for the Lifestyle Medicine movement in particular, as many who will seek care come armed with misinformation. A health care professional must be able to assess such fads and misinformation and discuss them with their patients in a productive manner as part of nutrition counseling.


*5. Understand when/how to refer patients to a qualified Registered Dietitian Nutritionist (RDN) or other professional and know the content of that consultation:*


Referral to a nutrition profession will depend on the credentials of the INTM nutrition graduate. Many will fall into this category themselves after passing the CNS exam. Others will have been working with colleagues in the Programs and become familiar with the strengths of those exclusively focused on nutrition counseling (e.g. rather than also performing primary care). In general, Integrative Medicine is about team-based medicine and knowing when and to what type of provider to refer a patient—I often refer to them as the quarterback of the patient care team. Further, they conduct peer review of every case study and patient care plan they create in this program, which allows them to experience the strength of an interdisciplinary team firsthand and to learn the strengths of various backgrounds. Thus, consultation in practice is second nature, and they are easily able to identify the appropriate specialty for such consultations. In fact, we strongly and actively encourage them to utilize their classmates as part of their clinical network after graduation.


*6. Recognize and plan for personal nutrition, physical activity, and wellness:*


This is the INTM nutrition curricula as well as Lifestyle Medicine in a nutshell. We emphasize “walking the talk” and have the students use themselves as their first patient in many instances. We also discuss how and when to use personal experience as part of nutrition counseling. The many discussions about their own experiences lead easily into discussions of our 40–60 cases in the program, which are very in depth and include many nuances that would likely be overlooked by a health care professional without nutrition education. The goal is for personal nutrition, physical activity, and wellness to be their automatic approach combined with the systems thinking and patient-centered nature of Integrative Medicine.

As Van Horn et al. explains, the idea that EPAs “promote individualized and personalized education” is especially fitting for the INTM nutrition curricula and Lifestyle Medicine, fitting in line with the goal provided by IAMSE. Further, the INTM nutrition curricula were designed to meet the competencies for the CNS exam and have been recognized as fulfilling the academic requirements for eligibility. The CNS competencies align well with Van Horn et al. and IAMSE with a greater emphasis on precision and personalized nutrition such as assessment of single nucleotide polymorphism (SNPs) and functional testing, e.g. hormonal testing [25]. Of note, is Competency 10. *Identification of symptoms that require medical referral* [25], which fits nicely with the idea of the INTM nutrition graduates acting as the quarterback of the health care team.

#### 3.3.3. Assessments

Assessments identified by Van Horn et al. and their application in the INTM nutrition curricula are as follows:


*1. Clinical rotations—direct observations:*


Our students are not required to have in-person experiences. However, we offer them the option of rotating with our Medical Director for 2 weeks. Further, the CNS exam requires 1000 h of supervised nutrition practice, which is extracurricular. This provides them the opportunity to get their feet wet with the oversight of a credentialed CNS or other nutrition health professional. This is a higher order experience than simply observing, especially given their extensive experience with case studies throughout the INTM nutrition curriculum. We have found that our students are well prepared for their 2 week rotation as part of our INTM Fellowship, which we attribute to their repetitive practice with numerous case studies followed by peer review of various approaches.


*2. Standardized patients—360° evaluations:*


While we do not have in-person standardized patients, we do attempt to replicate this with our numerous case studies. Further, we utilize discussions of real world issues throughout the courses, including issues that students are currently seeing in their practice.


*3. Case-based conferences—chart-stimulated recall:*


This may be conducted during in-person experiences (optional/extracurricular) but is not formally a part of the INTM nutrition curricula outside of the INTM Fellowship, at least in its purest form. However, our online modules utilize a similar approach (see below).


*4. Online modules—chart audit:*


The INTM nutrition curricula is built around online modules. When the students discuss different approaches to the same cases, this is similar to a chart audit. This type of discussion is often on a weekly basis and serves as the most clinically meaningful experiences in the curricula—each is essentially a dry run for their practice. This is like a combination of a consult with colleagues (emphasizing team based medicine and referrals) and the standard morbidity and mortality (M&M) conference, including learning from each other’s missteps and different perspectives.


*5. National courses—in-training examination, patient surveys:*


This is where the accrediting body comes into play. For the INTM nutrition curricula, this is the Board for Certification of Nutrition Specialists. They provide the CNS exam as well as continuing education requirements and events. However, it is paramount the opportunities for continuing education continue to grow. As such, GW INTM strives to provide continuing education for all health care providers in the fields of nutrition and related Integrative and Lifestyle Medicine topics. Varied, robust opportunities for continuing education will be the key to growing the field and maintaining up-to-date knowledge. This includes the use of non-traditional methods such as webinars and on-demand, online modules. Continuing education events also offer the opportunity to embark on nutrition training without the commitment of a degree program.

#### 3.3.4. Concerns of Note

When it comes to nutrition education in general, my biggest concern is prior knowledge—pre-conceptions from prior experience and the culture—that may affect the ability of the learner to learn [26]. One must consider what the learner is bringing into the classroom with them including emotions and previous knowledge and experiences [26]. These must be challenged or reaffirmed to encourage transformative learning—demolishing the current scaffolding in favor of new knowledge. To successfully shift the learner’s perspective in this manner, we must give them the opportunity to reflect on their own understanding in the face of new information or a new perspective.

In the GW INTM Programs, there is a broad range of prior knowledge. There are some students who are pre-medicine, who may be transitioning from another field or health care profession. Then there are clinicians who have been in practice for decades. What we can assume is that they have not had these materials presented to them in the same way as the INTM nutrition curricula. The Integrative and Lifestyle Medicine approach is unique, and that is what they are seeking. More importantly, the content we are presenting may be correction of an outdated understanding or even misinformation. Misinformation is particularly widespread and problematic in nutrition due to single study sensationalism, misrepresentation in various media outlets including social media and pop-science from so-called experts, societal beliefs, etc. While this prior knowledge may be working against us, the students are enrolled in the INTM nutrition concentration because they know they have insufficient understanding in this area, meaning they are open-minded to some extent at least. Given the wide range of prior knowledge, spending significant efforts on identification of these errors is not overly valuable. Instead, the focus needs to be on facilitating transformative learning.

This approach is supported by Van Horn et al. as is apparent from the following excerpt:

“Medical nutrition educators teach solid foundations and current guidelines while preparing learners for new concepts just over the horizon. They accomplish this by *strengthening critical thinking skills*, *encouraging questions on both old and new claims*, and *illustrating the importance of lifelong learning*. [emphasis added]” [19].

One of the strengths of the INTM Programs is that they are interprofessional education. As reported in Thomas, Kern, Hughes, & Chen [17], the World Health Organization and the US National Academy of Medicine (formerly the Institute of Medicine) are encouraging the use and expansion of interprofessional education as a means for the improvement of health and patient outcomes. Utilization of the guidelines surrounding interprofessional education could further strengthen this.

### 3.4. A Nutrition Education Course: Nutritional Immunology

The final course for the INTM nutrition curriculum is Nutritional Immunology—a synthesis and application of the knowledge and skills acquired through the previous coursework. This is a real world application at the highest level of complexity, which assesses the ability of the learner to function in the worst case scenario in a way that is also applicable to daily clinical practice. Thus, while it is extreme, it is not unrealistic. The University approved learning objectives for this course follow.

As a result of completing this course, the student will be able to:Distinguish the nutrients that are involved in the function and regulation of the immune system. (Cognitive: Knowledge)Examine the interaction between the immune system and nutrition to develop effective, personalized, culturally-appropriate Integrative Nutrition Care Plans that support immune function. (Psychomotor: Skill and Behavioral)Develop effective, personalized, culturally-appropriate Integrative Nutrition Care Plans that support immune function. (Psychomotor: Skill and Behavioral)

In this course, we need to (re)introduce the students to immunology, move with the students from nutrient to nutrient while assuring they retain the understandings from previous weeks, and then that they integrate this comprehensive understanding into an oral presentation (final project) and ultimately utilize this in patient care (application to practice). The key is to focus on the most important nutrients and then to scaffold each week to maintain understanding throughout the course as well as to identify themes. To promote integration and retention, we use principles of adult learning emphasizing active learning including discussion boards & cases and peer review [27].

While much of the course focuses on Western cultures (as this is who Integrative and Lifestyle Medicine providers are treating in the United States), many topics lend themselves to an international health and public health discussion. For instance, with vitamin A, we discuss night blindness and that supplementation reduces all cause mortality in developing countries. We also spend time on the acute phase response and the role of chronic inflammation—from chronic disease or endemic infections—in reducing apparent nutritional status. We discuss how to calculate true vitamin A and iron status in such cases.

#### Educational Strategies

The emphasis on understanding one’s own diet and nutrition is included in our new nutrition curricula. We typically have them start with their own experiences and then progress to case studies. This allows them to understand the patient’s perspective better. One could also imagine that a busy provider may not take the time to apply their newfound nutrition knowledge to their own life if the time is not expressly given for this purpose. This reinforces them “walking the talk” as well.

As discussed previously, nutrition knowledge of physicians correlated with their personal behaviors. By improving each of our student’s ability to care for themselves we hope that this will better equip them to care for others. The potential for decreased burnout or other negative outcomes as a practitioner who cares for themselves is also central to our approach. For instance, our program includes a 2-credit course, Self Care Methods for Health Care Professionals, in which each student prepares a self care plan addressing the Eight Dimensions of Wellness.

The Nutritional Immunology course does not place emphasis on the student’s personal nutrition or diet, however. This is in large-part because this is their last class, and, thus, the emphasis is placed on the application of their knowledge to patient care. While there is occasion to give the student time to think about nutrition in their own life, I am not sure this should be the emphasis of this final course. However, this may be an opportunity to tie in reflection by having them apply what they have learned to their own lifestyles and reflecting upon how their approach has changed over the progression of the course. Practically, this could be done by having them reflect each week in a reflection journal and then synthesize this into a short reflection paper at the conclusion of the course.

Developing a nutrition care plan is a psychomotor-skill/competence objective. What is best suited to this is real life clinical experience, simulation, role playing, standard patients, audiovisual review of learner, and perhaps demonstration. In the online environment, we rely on the learner playing the role of a clinician through case study analysis and care plan creation. Then we have them post this for discussion, which helps reinforce the cognitive/knowledge component and move into the affective/attitudinal component. Thus, we address different types of learners as well as different levels of objectives.

In Nutritional Immunology, we address the real life clinical experience through role playing in our discussion board assignments (psychomotor/competence). In these assignments, the students will apply their learning to-date to a case study in which they are asked to develop a nutrition care plan. To reinforce the cognitive/knowledge component and move into the affective/attitudinal component, we have the students post their nutrition care plans to the discussion boards and discuss the various different approaches and their pro’s and con’s—similar to a consult or M&M conference. In this same vein, the learner selects an area to take a deep dive and create a 10-minute oral presentation, which hones their communication skills, emphasizing information synthesis and concise communication of complex topics. Each learner also conducts peer review of two oral presentations, which utilizes critical thinking and practices giving effective feedback. This also strengthens the interprofessional bonds and their recognition of the value of team-based medicine. In combination, this is an application of the “see one, do one, teach one” concept [28].

## 4. Conclusions

As one of six pillars of Lifestyle Medicine, nutrition is a foundation of health; however nutrition education must be improved for Lifestyle Medicine, or an approach inclusive of lifestyle, to truly become the foundation of health care. Right now, Lifestyle Medicine is more a symptom of poor nutrition training for physicians and other non-RD health care providers. The need for nutrition as the foundation of health care is now widely recognized by health care professionals and the public. Despite appreciating the importance of nutrition in general and in chronic diseases, clinicians seldom counsel their patients on nutrition. As discussed in this article, this is largely due to inadequate or lack of training, leaving a significant need in patient care; a gap that can be closed with evidence-based curricula in medical schools and in the trainings of other health care professionals. The George Washington Integrative Medicine Nutrition Program is a model of how pre- through post-professional health care providers may be equipped with the nutrition counseling skills necessary to tackle nutrition with their patients as first-line therapy. This includes utilization of team-based medicine and referrals to more specialized health care professionals such as RDs.

## Figures and Tables

**Table 1 ijerph-18-05974-t001:** Comparison of the Approaches of Van Horn et al. and Thomas et al.

Van Horn et al. 2019	Thomas et al. 2015
Define terms, scope, and application;	Step 1
Identify and engage diverse stakeholders;	Step 1 & 2
Collect data;	Step 1 & 2
Draft nutrition competencies for stakeholder review;	Step 3
Apply the competencies, e.g., Curricular Design, Process Improvement, Program Evaluation;	Step 4, 5, & 6
Periodically review and provide updates.	Step 6

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
