# Peer review of "Nutrition, a Tenet of Lifestyle Medicine but Not Medicine?"

_ijerph, 2021, doi:10.3390/ijerph18115974_

Round 1

Reviewer 1 Report

The author was responsive to reviewer comments. Changes made in particular the introduction paragraph and subheadings have made this manuscript easier to follow.

Author Response

Thank you for your review.

Reviewer 2 Report

I encourage authors resubmission, yet, i belive that the manuscript itself do not provide anyhting new to the scientific comunity. 

My coments and suggestions are still the same from R1 version of the manuscript. 

Round 2

I encourage authors resubmission, yet, i belive that the manuscript itself do not provide anyhting new to the scientific comunity.

My coments and suggestions are still the same from R1 version of the manuscript.

Response: Perhaps this is somewhat of a country by country issue. In the United States, the use of nutrition in medical care is rare; and, thus, such an approach, perhaps as a part of Lifestyle Medicine, is novel. The argument for such an approach and how to prepare our clinicians is necessary if not novel. Without such an argument, we cannot make in-roads to clinician education—a problem I myself am actively facing. While I have been able to build this post-professional education program, we still cannot educate our medical students. This is why this article is crucial to the advancement of the field. I have added additional detail to make this clearer.

In addition, I have gone through your original comments for a second time. I hope that I have clarified any remaining issues. Thank you.

Round 1

Thank you for the opportunity to review this manuscript. Raising awareness about the importance of and need for training in nutrition within medical education is imperative. As written the aims of the paper are unclear. The paper is prepared as a review, but appears to be more focused as a case study (with an introduction serving as a review of the literature) at George Washington School of Medicine and Health Sciences where the Van Horn et al. competencies have been applied. Based on the manuscript as written strengths, weaknesses and specific comments have been highlighted.

Strengths:

The introduction demonstrates the need for nutrition training in lifestyle medicine (medical education).

Weaknesses:

It may be helpful to organize the introduction in a manner that aids the reader through the available information on this topic. Consider presenting information chronologically. As written the information is shared about guidelines from professional organizations, contributions of individual studies/reviews and the NHLBI workshop. What is most important to share to support your study purpose? Given the substantial focus on the Van Horn et al. work perhaps that should be the focus as it also then aligns with the case study?

The Case Study is a great demonstration of how the suggested Van Horn et al. competencies can be applied, but comes as a surprise given the lack of specific mention in the abstract and/or study aims.

Response: Thank you very much for the feedback. I added an introduction to introduce the reader to the structure and purpose of the article as well as headers, which I had hoped would help guide the reader more throughout the article. Since that is still not clear, I have extended the introduction and have detailed the aims in this. I have also included the argument for why this article is important. Further, it is already in chronological order, but I have added headers to make the topics clear. Thank you.

Greater detail is needed in-regards-to how the INTM nutrition curriculum is meeting the competencies. Current information is vague and does not provide specific examples indicating the INTM curriculum is meeting the outline criteria as intended.

Line 176: What kind of nutritional assessment is included within the assessment of the patient planning process? The response to this competencies does now show nutrition assessment is being learned.

Response: I erred on the side of brevity but have fleshed this out at your suggestion. Thank you.

It may be helpful to include a nutrition professional as an author on this manuscript to ensure appropriate interpretation of nutrition-related competencies.

Response: That is actually my background, so I agree!

Throughout the manuscript many personal thoughts/perceptions are included and should be removed (e.g., Line 203, Line 255, etc.)

Response: This is a stylistic choice that I would like to retain.

Specific comments:

Line 29: Consider removing the line “However, 60% are not receiving nutrition counseling” given the lack of citation and the subsequent sentence provides data on the estimated primary care visits with nutrition counseling.

Response: I mis-assumed this would be clear. I have added the citation to this line as well. Thank you.

Lines 125 – 137: Using a blog post (citation 24) is less than ideal given the lack of peer review for this type of citation.

Response: I agree. But alas, this is how the IAMSE has chosen to communicate their findings to-date. It is better to include the currently available information than no information at all.

Lines 139 – 162: Consider organizing this information with subheadings or within a table to more clearly show what is required/offered on a curriculum level (e.g., Masters, concentrations, etc.) currently and what is planned for the future.

Response: I have added headers throughout and hope this makes it clearer and easier to navigate. Thank you.

Line 183: How is the immunology course related to competency #2 that is focused on counseling of key chronic diseases?

Response: I think you are referring to competency #3: Counsel patients on basic public health nutrition issues (e.g., obesity prevention and treatment, hypertension, cardiovascular disease, and diabetes)
I discuss this in a later section (3.4) and have referenced it here for added clarity. Thank you,

Lines 290 – 312: Why is the immunology course a separate section? It is mentioned within the case study and may be more appropriate in that location given it appears to be part of the INTM curriculum.

Response: I agree now that I see it with fresh eyes and have updated the headings to reflect this. Thank you for the suggestion.

Lines 314 – 319: What is the evidence for using this type of approach?

Response: This is related to work discussed at the beginning, which I am now referring back to in addition to adding more detail about the logic behind the approach. Thank you.

Line 351-352: Include citations to support this statement.

Response: This was extensively covered at the beginning of the manuscript. I now reference that here. Thank you.

Reviewer 3 Report

The paper is based on the premise that health professionals in the United States lack nutritional knowledge, and that this knowledge is of relevance in clinical are (both in-patient and out-patient). Based on this premise, the author argues for “Lifestyle Medicine” and describes a case study of nutritional education for health professionals, based on frameworks from Van Horn et al. (2019)  and Thomas et al. (2015). The article is well-written and the nutritional education is described in detail.

However, there are problems with some fundamentals of the article. It is very normative and seems more like promotion of Lifestyle Medicine than a scientific examination of it. As the argument flows, there is quite a leap of logic that such promotion of Lifestyle Medicine's particular philosophy (rather than conventional dietetics) somehow follows from the premise that health professionals in the United States lack nutritional knowledge. There is no good argument for Lifestyle Medicine in particular.

This is a specific movement with a particular philosophy. But there is already a profession with expertise in dietetics: dietitians. You mention "the traditional Registered Dietitian" on line 184 and discuss referral to a nutrition professional (e.g. a dietitian) on point 5 under “Competencies”. But it seems as dietitians are only considered in clinical treatments of patients but not on in prevention? I don’t know about the United States but this is not true for the countries I know about.

So my main issue is that the paper seems to build on the premise that Lifestyle Medicine has something to offer that established dietetics doesn’t, but this is yet unconvincing. I would re-frame the argument with Lifestyle Medicine as one of several potential ways of forming evidence-based nutritional education. But give more credit to already-established ways of doing it.

A detail: An opinion piece by Neal Barnard is mentioned, and I fail to see why. He is not an authority, and even if he was an opinion piece has no value as evidence. I would delete that part and instead provide evidence for why Lifestyle Medicine would be superior to established dietetics.

Author Response

The paper is based on the premise that health professionals in the United States lack nutritional knowledge, and that this knowledge is of relevance in clinical are (both in-patient and out-patient). Based on this premise, the author argues for “Lifestyle Medicine” and describes a case study of nutritional education for health professionals, based on frameworks from Van Horn et al. (2019)  and Thomas et al. (2015). The article is well-written and the nutritional education is described in detail.

However, there are problems with some fundamentals of the article. It is very normative and seems more like promotion of Lifestyle Medicine than a scientific examination of it. As the argument flows, there is quite a leap of logic that such promotion of Lifestyle Medicine's particular philosophy (rather than conventional dietetics) somehow follows from the premise that health professionals in the United States lack nutritional knowledge. There is no good argument for Lifestyle Medicine in particular.

This is a specific movement with a particular philosophy. But there is already a profession with expertise in dietetics: dietitians. You mention "the traditional Registered Dietitian" on line 184 and discuss referral to a nutrition professional (e.g. a dietitian) on point 5 under “Competencies”. But it seems as dietitians are only considered in clinical treatments of patients but not on in prevention? I don’t know about the United States but this is not true for the countries I know about.

So my main issue is that the paper seems to build on the premise that Lifestyle Medicine has something to offer that established dietetics doesn’t, but this is yet unconvincing. I would re-frame the argument with Lifestyle Medicine as one of several potential ways of forming evidence-based nutritional education. But give more credit to already-established ways of doing it.

A detail: An opinion piece by Neal Barnard is mentioned, and I fail to see why. He is not an authority, and even if he was an opinion piece has no value as evidence. I would delete that part and instead provide evidence for why Lifestyle Medicine would be superior to established dietetics.

Response: Thank you for the feedback. It was certainly not my intention to slight RDs at all, and I am glad that you made me aware that it was coming off that way. I was really focused on everyone else and their lack of nutrition training including knowledge of when to refer. I hope that my edits have made that clear now.

Further, I have provided additional context for the article. It is not an argument for Lifestyle Medicine; rather, LM is the argument for the need for nutrition education for all health care providers. In order to change the current status quo, of nutrition not being included in medical education or health care professional education at large, the case must be made for this change. This is the goal of this manuscript.

I have also removed the mention of a literature review, as this is merely an introduction/background. Thank you.

Reviewer 4 Report

This is intended to be a narrative review about how Nutrition training can be introduced in Medical education as a first step to increase the knowledge about the importance of lifestyle medicine in the prevention and treatment of several chronic diseases and longevity.

The main comment is about the purpose of the article. As the author mentions in the introduction, “A review of the literature will be followed….” , but, there is no information about the methodology to conduct such review. Even being a narrative review, a brief explanation about the sources of information and the selection process should be explained. Besides the analysis and the case study are both focused on US institutions with no reference to other countries of the Americas, Europe, Oceania or other non-US universities. It will improve the interest of the article if the author explores how the nutrition principles are integrated into the medical school curriculum in countries around the world. Otherwise, the author should indicate that the review and the case study refer to the US.

Another aspect that should be addressed is about one of the most common complains of primary care physicians around the world when asked about the difficulties in incorporating diet and physical activity counselling into their daily practice, namely short length consultation time. Several studies have reported that one of the most common barriers to provide nutrition counselling in general practice is lack of time.  It would be interesting to provide guidance about how to incorporate nutrition counselling and health promotion into the physician consultation without stressing even more the daily medical consultation, which is quite common in general practice (at least in the European countries). It would be also interesting to discuss the role Registered dietitians have to support other health care professionals providing nutrition education.

Minor comments

Line 30, the author should mention the importance of physical activity, in addition to avoiding sedentary behaviors, in the sentence “…use of tobacco, sedentary behavior, …..”

 Line 35, Use the word Diet instead of nutrition in the sentence “ Nutrition and other lifestyle interventions…..” Nutrition is a physiological process, diet is a behavior which can be modified.

Line 58. Add a reference to the sentence.

Line 68. In addition to lack of knowledge or confidence, one of the most common barriers to provide nutrition counselling is lack of time, the author should add also a reference to the sentence.

Line 121. The author should explain why the Thomas method is the reference to be used. Readers who are not expert on the education area may need a brief explanation about the model.

Line 160, the author should clarify what “pure nutrition” means.

Author Response

This is intended to be a narrative review about how Nutrition training can be introduced in Medical education as a first step to increase the knowledge about the importance of lifestyle medicine in the prevention and treatment of several chronic diseases and longevity.

The main comment is about the purpose of the article. As the author mentions in the introduction, “A review of the literature will be followed….” , but, there is no information about the methodology to conduct such review. Even being a narrative review, a brief explanation about the sources of information and the selection process should be explained. Besides the analysis and the case study are both focused on US institutions with no reference to other countries of the Americas, Europe, Oceania or other non-US universities. It will improve the interest of the article if the author explores how the nutrition principles are integrated into the medical school curriculum in countries around the world. Otherwise, the author should indicate that the review and the case study refer to the US.

Response: Thank you for your review and perspective. I have removed the mention of a literature review, as this is merely an introduction/background. I have also clarified the locations being referenced.

Another aspect that should be addressed is about one of the most common complains of primary care physicians around the world when asked about the difficulties in incorporating diet and physical activity counselling into their daily practice, namely short length consultation time. Several studies have reported that one of the most common barriers to provide nutrition counselling in general practice is lack of time.  It would be interesting to provide guidance about how to incorporate nutrition counselling and health promotion into the physician consultation without stressing even more the daily medical consultation, which is quite common in general practice (at least in the European countries). It would be also interesting to discuss the role Registered dietitians have to support other health care professionals providing nutrition education.

Line 68. In addition to lack of knowledge or confidence, one of the most common barriers to provide nutrition counselling is lack of time, the author should add also a reference to the sentence.

Response: I could not agree more on the importance of this topic. In fact, it warrants its own manuscript. It is, however, outside the scope of this article. I am not addressing barriers in practice, but rather the lack of training--one barrier at a time when it is such a fundamental issue. Further, my expertise here is in education. Thank you.

Minor comments

Line 30, the author should mention the importance of physical activity, in addition to avoiding sedentary behaviors, in the sentence “…use of tobacco, sedentary behavior, …..”

Response: Since this is phrased in a negative light (what to avoid), mentioning physical activity does not make sense unless the entire sentence was rephrased. It should be clear that avoiding sedentary behavior requires increase physical activity, though. Thank you.

 Line 35, Use the word Diet instead of nutrition in the sentence “ Nutrition and other lifestyle interventions…..” Nutrition is a physiological process, diet is a behavior which can be modified.

Response: I hear what you are saying. This is not an understanding of clinicians writ large, though. Thus, I tried to maintain the use of nutrition as a field throughout instead of moving from nutrition and diet, which would potentially decrease clarity to a broader audience. I have gone through and double-checked that this has been maintained throughout. Thank you.

Line 58. Add a reference to the sentence.

Response: Added. Thank you.

Line 121. The author should explain why the Thomas method is the reference to be used. Readers who are not expert on the education area may need a brief explanation about the model.

Response: This is the typical method used in US medical school. I have added this explanation. Thank you.

Line 160, the author should clarify what “pure nutrition” means.

Response: The adjective was unnecessary and has been deleted. Thank you.

Round 2

Reviewer 3 Report

Thank you for your revised paper. I feel like you have reformulated the paper in a way that sufficiently present LM more as a philosophy among others and no longer as superior. I also think you sufficiently address RDs.

Just for the record, I would like to repeat that I do not consider Neal Barnard's opinions worthy of citation. That said, this is my opinion and my job as a referee is not to be a gatekeeper of what my colleagues should be "allowed to say" in papers. I will therefore not insist on changing this, we simply disagree on this issue (which is fine).

I only have ONE suggestion now. Since it is now categorized as "Communication" then perhaps "This narrative review presents" (line 11) should be reformulated to "This communication piece presents", or something similar?

Thank you again for your revision.

Author Response

Very good point. Thank you for catching that. I have changed this to now read: "This communication presents..."

This manuscript is a resubmission of an earlier submission. The following is a list of the peer review reports and author responses from that submission.

Round 1

Reviewer 1 Report

This scientific article presents a clear and concise summary of the current means and the efforts carried out by different organizations and institutions to try to combine current knowledge in relation to nutrition in the medical and clinical environment.

In the article there are some errors related to the text format, leaving unnecessary spaces, check this aspect.

Too many keywords, there should be a maximum of 5 or 6.

It must be at some point reflected what is the objective of the study, and justified this in the introduction.

If it is a review article, the search, inclusion and exclusion criteria of this article must be indicated.

At the end of the article, despite being highly interesting, I personally find it more of an informative article and for purposes unrelated to the scentific comunity, but for the benefit of the ACLM association.

Reviewer 2 Report

Thank you for the opportunity to review this manuscript. Raising awareness about the importance of and need for training in nutrition within medical education is imperative. As written the aims of the paper are unclear. The paper is prepared as a review, but appears to be more focused as a case study (with an introduction serving as a review of the literature) at George Washington School of Medicine and Health Sciences where the Van Horn et al. competencies have been applied. Based on the manuscript as written strengths, weaknesses and specific comments have been highlighted.

Strengths:

  • The introduction demonstrates the need for nutrition training in lifestyle medicine (medical education).

Weaknesses:

  • It may be helpful to organize the introduction in a manner that aids the reader through the available information on this topic. Consider presenting information chronologically. As written the information is shared about guidelines from professional organizations, contributions of individual studies/reviews and the NHLBI workshop. What is most important to share to support your study purpose? Given the substantial focus on the Van Horn et al. work perhaps that should be the focus as it also then aligns with the case study?
  • The Case Study is a great demonstration of how the suggested Van Horn et al. competencies can be applied, but comes as a surprise given the lack of specific mention in the abstract and/or study aims.
  • Greater detail is needed in-regards-to how the INTM nutrition curriculum is meeting the competencies. Current information is vague and does not provide specific examples indicating the INTM curriculum is meeting the outline criteria as intended.
  • It may be helpful to include a nutrition professional as an author on this manuscript to ensure appropriate interpretation of nutrition-related competencies.
  • Throughout the manuscript many personal thoughts/perceptions are included and should be removed (e.g., Line 203, Line 255, etc.)

Specific comments:

  • Line 29: Consider removing the line “However, 60% are not receiving nutrition counseling” given the lack of citation and the subsequent sentence provides data on the estimated primary care visits with nutrition counseling.
  • Lines 125 – 137: Using a blog post (citation 24) is less than ideal given the lack of peer review for this type of citation.
  • Lines 139 – 162: Consider organizing this information with subheadings or within a table to more clearly show what is required/offered on a curriculum level (e.g., Masters, concentrations, etc.) currently and what is planned for the future.
  • Line 176: What kind of nutritional assessment is included within the assessment of the patient planning process? The response to this competencies does now show nutrition assessment is being learned.
  • Line 183: How is the immunology course related to competency #2 that is focused on counseling of key chronic diseases?
  • Lines 290 – 312: Why is the immunology course a separate section? It is mentioned within the case study and may be more appropriate in that location given it appears to be part of the INTM curriculum.
  • Lines 314 – 319: What is the evidence for using this type of approach?
  • Line 351-352: Include citations to support this statement.

Reviewer 3 Report

The present manuscript entitled "Nutrition, A Tenet of Lifestyle Medicine But Not Medicine?" is an article in which it is observed that nutricion and lifestyle medicine is one of the most important thigs to prevent chronic desises. Medical education in lifestyle medicine is, therefore, proposed as a necessary intervention to allow all health providers to learn how to effectively and efficiently counsel their patients toward adopting and sustaining healthier behaviors. However, in my opinion, this article does not provide relevant knowledge to science.